# Research Progress on the Dynamic Characteristics of Planetary Gear Transmission in a Non-Inertial System

**Bingwei Gao [1,2,\*], Yongkang Wang [1,2] and Guangbin Yu [3]**

1   Key Laboratory of Advanced Manufacturing and Intelligent Technology, Ministry of Education, Harbin University of Science and Technology, Harbin 150080, China; 2220100056@stu.hrbust.edu.cn
2   School of Mechanical and Power Engineering, Harbin University of Science and Technology, Harbin 150080, China
3   School of Mechatronics Engineering, Harbin Institute of Technology, Harbin 150006, China; yugb@hit.edu.cn
\*   Correspondence: gaobingwei@hrbust.edu.cn; Tel.: 86-0451-86390588

**Abstract:** Planetary gear systems have many advantages over other gear systems. Previous studies on its dynamic characteristics mostly used Earth as the reference system, which is inconsistent with the actual working conditions of many planetary gear systems, such as aircraft maneuvering, vehicle movement changes, etc. By analyzing representative research papers, this study summarizes the lumped-parameter, finite element, and rigid–flexible coupling models commonly used in studying the traditional dynamic characteristics. Then, the research status of gear–rotor and planetary gear systems in inertial and non-inertial systems is summarized. The research progress of load characteristics, vibration characteristics, and vibration control of the traditional planetary gear system is summarized. Finally, some suggestions for future development are put forward. There are a few studies on the non-inertial dynamics of planetary gear systems. The three analysis models have distinct characteristics and applications but can all be used in non-inertial systems. The dynamic analysis method of non-inertial rotor systems can be combined with the dynamic study of gear systems. It is of practical significance to study the non-inertial dynamic characteristics of planetary gear systems. Scholars can refer to the non-inertial dynamic research of the gear–rotor system, select the analysis model according to the needs, and continue to study the dynamic characteristics of the planetary gear system under the non-inertial system.

**Keywords:** planetary gear system; dynamic model; non-inertial dynamics; dynamic characteristics; vibration control

## 1. Introduction

The gear system is the most widely used transmission form at present. Among them, the planetary transmission system is widely used in aerospace, wind power, automobile, shipbuilding, and in the chemical industry and other fields, due to its advantages of a compact structure, a large transmission ratio, high reliability, and high transmission efficiency. Figure 1 is a schematic diagram of the principle of a planetary gear transmission system for wind power [1]. The performance requirements of planetary gear systems are constantly increasing in various application fields, especially in the field of aviation. Exploring its dynamic characteristics has practical significance for vibration and noise control, dynamic load calculation, and lightweight design. Based on both the vibration theory of gear transmission systems and the dynamic systems theory [2], relevant scholars have conducted extensive and in-depth research from the perspectives of incentive factors, dynamic model construction, model solution, inherent characteristics, dynamic characteristics, etc., and achieved many results.

The so-called inertial system refers to Earth as the reference system. At present, the main research is on the dynamic characteristics of planetary gears in the inertial reference frame. But, in actual works, the transmission system will move together with aircrafts,

ships, and other carriers. We call the mechanical conditions in this kind of motion environment "the non-inertial system" [3]. Figure 2 shows the schematic diagram of helicopter maneuvering [4]. In non-inertial systems, Newton's "classical mechanics" no longer applies. A hypothetical "inertial force" needs to be introduced to study the motion and dynamics in non-inertial systems. For gear transmission systems working under extreme conditions such as high speed, heavy load, and high vibration, when the body changes its motion state rapidly, the non-inertial effect will cause changes in absolute acceleration and gravity components. This will directly affect the behavior dynamics of the planetary gear system [5]. Therefore, studying the dynamic characteristics of planetary gears in non-inertial systems is necessary.

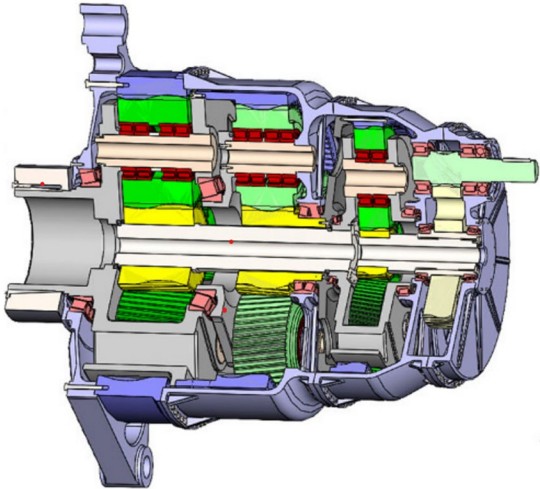

**Figure 1.** Schematic diagram of planetary gear system for wind power [1].

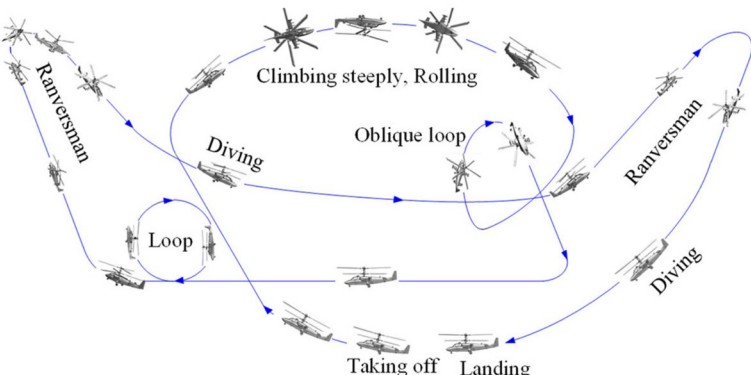

**Figure 2.** Helicopter maneuvering diagram [4].

Research on the dynamic characteristics of planetary gear systems has always been a hot issue in this field. Parker et al. [6] reviewed the dynamic mathematical model, vibration mode characteristics, and dynamic response of the planetary gear system. In 2019, Zghal et al. [7] established a first-stage planetary gear train translation–torsion lumped-mass model in the stationary coordinate system. However, the structure of the planetary gear system is more complex and the working condition is worse than that of the fixed shaft gear system. There are still many difficulties in exploring its dynamic characteristics, fault prediction, vibration and noise control, etc. Moreover, most of the current research is based on the inertial system. Scholars often ignore the influence of non-inertial effects. This does not meet the requirements of high-maneuvering flight and other sports environments with rapid acceleration changes.

In short, it is found that the planetary gear transmission system is widely used in engineering practice. This includes a variety of non-inertial environments. Scholars have found

in the research on the non-inertial rotor system that the system's dynamic characteristics in the non-inertial environment are quite different from those in the inertial system. For example, the additional load generated during the maneuvering flight will directly affect the system's normal operation. In order to further improve the transmission performance of the planetary gear system, it is necessary to consider the influence of non-inertial factors. However, there are few studies on the dynamic characteristics of gear transmission systems in non-inertial systems.

This paper first summarizes the research models of the dynamic characteristics of planetary gears. The characteristics and selection principles of the three models are obtained by comparing the parameter concentration model, the finite element model, and the rigid–flexible coupling model. Subsequently, the basic content of non-inertial dynamics is briefly described and the current research results of gear–rotor systems and planetary gear systems in non-inertial frames are analyzed. The connection between the two research methods of the system's dynamic characteristics is obtained, which provides a reference for the research on the dynamics of the planetary gear system in the non-inertial system. Then, according to the existing research results of the dynamic characteristics of the planetary gear system in the non-inertial and inertial system, the dynamic load and load-sharing characteristics, vibration characteristics, and vibration control research statuses are summarized. Finally, the future development direction of the dynamic characteristics of the planetary gear train is discussed, which provides research ideas for relevant researchers.

## 2. Dynamic Models

The establishment of a dynamic model is the basis for the study of dynamic characteristics. The research on the dynamic characteristics of planetary gear systems still belongs in essence to the research category of gear dynamics. Therefore, its dynamic model has also experienced development from the linear model to the non-linear model, from the time-invariant model to the time-varying model, and from the single-stage system to the multi-stage system. At present, the factors that are often considered when establishing the dynamic model of the planetary gear system are time-varying mesh stiffness, backlash, the gyro effect, comprehensive transmission error, meshing force, support conditions, structural flexibility, etc. It can be seen that the establishment of analytical models, considering non-linear and time-varying factors, has gradually become the mainstream of research. According to the method used and the factors considered in establishing the dynamic model, the model can be divided into the lumped-parameter model, the finite element model, the rigid–flexible coupling model, and so on. In this section, by analyzing the application status of the three models, their advantages, disadvantages, and application occasions are obtained. It is discussed whether the traditional dynamic model is suitable for the non-inertial system.

### 2.1. Lumped-Parameter Model

The lumped-parameter model is used to simplify the components of the planetary gear transmission system by using the idea of centralized parameters to obtain the mass point with the concentrated mass and the moment of inertia. Dynamical models are constructed using virtual springs and dampers as a link between the motion and spatial relationships and the mass points. Scholars have verified the validity and accuracy of the model. It is mostly used to analyze the natural frequency, dynamic response, and qualitative analysis of the motion of planetary gear systems with simple structures.

Li et al. [8] explained in detail the lumped-parameter model of a gear transmission system. In 1999, Paker [9] established the bending–torsion coupling dynamic model of a single-stage straight planetary gear train by using the lumped-mass method and studied the system's natural frequency when the planetary gear was uniformly distributed. In 2001, Kahraman [10] established the pure torsion dynamic model of a single-stage planetary gear transmission system and analyzed the inherent characteristics of the model. In 2007, Al-Shyyba [11] established the torsional vibration model of the planetary gear transmission

system based on the discrete non-linear idea and the system's dynamic response was obtained using the harmonic balance method. In 2007, Ambarisha et al. [12] studied the non-linear dynamics of planetary gear transmission under meshing stiffness excitation by using the lumped-parameter model. In 2010, Guo et al. [13] established the purely torsional dynamic model of the compound planetary gear transmission system and studied the influence of the natural frequency and stiffness parameters of the system on the vibration characteristics. Zhu et al. [14] established a 2K-H planetary gear translation–torsion coupled non-linear dynamic model considering friction, time-varying mesh stiffness, tooth backlash, and comprehensive meshing error. In 2011, Li et al. [15] established a pure torsion dynamic model of the planetary gear system considering time-varying mesh stiffness, comprehensive meshing error, and non-linear factors of tooth backlash. In 2012, Gu [16] proposed a lumped-parameter model of the planetary gear train considering the position error of planetary gears. In 2013, Cooley et al. [17] established the planetary gear dynamic model considering the gyro effect based on the set-parameter method and gave three vibration modes of the system. Sandkar et al. [18] established a bending–torsion–shaft–swing dynamic model of a herringbone planetary gear train. The model has up to 126 degrees of freedom, but its solution process is very complicated. In 2016, Ahmed et al. [19] established a torsion lumped-parameter model of the planetary gear system with a variable load. The numerical results were consistent with those obtained from the planetary gear test bed.

In 2017, Zhang et al. [20] established a three-dimensional dynamic model of a two-stage helical planetary gear by using the lumped-parameter method, as shown in Figure 3. The model was adapted for a different number of planets and different planetary phasing and spacing configurations in the two phases. In Figure 3, the component bearings, gear meshes, and interactions between the two stages are all modeled using linear springs (*k*). The damping is not shown in the model. In 2019, Liu et al. [21] proposed a two-stage planetary gear system with sliding friction and elastic continuous ring gear. The modal characteristics and resonance problems of the system are studied using the lumped-parameter method. MO et al. [22] established the dynamic model of the herringbone planetary transmission system based on the idea of parameter centralization and the Lagrange method. In 2020, Shen et al. [23] established a purely torsional vibration dynamic model of the 2K-H planetary gear set (Figure 4.). In the figure, *s* is the sun gear, *p* is the planetary gear, *c* is the planet carrier, and *r* is the ring gear. The research shows that the model can well reflect the influence of tooth surface wear on the dynamic characteristics. Zhang et al. [24] established a planetary gear train translation–torsion dynamic model using the lumped-parameter method. The natural frequency and vibration mode characteristics of the coaxial helicopter main reducer were analyzed. Leng et al. [25] established a translation–torsion coupling model of a two-stage planetary gear transmission system. The effects of the chaotic frequency and tooth backlash on the non-linear characteristics of a two-stage planetary gear train were studied. In 2022, Zhang et al. [26] established a purely torsional non-linear dynamic model of the herringbone planetary gear transmission system using the lumped-parameter method.

Through the above research, it can be found that the lumped-parameter model is widely used to study the dynamic characteristics of planetary gear systems. The lumped-parameter model has many advantages, such as simple structure, rapid modeling, convenient solution, and so on. As the number of factors considered in modeling gradually increases, and as most of the influencing factors are non-linear and time-varying, the complexity of the model sharply increases. The system was greatly simplified when building the model. The model often contains fewer degrees of freedom, so it is difficult to reflect the actual dynamic characteristics of the system accurately. Therefore, researchers can comprehensively consider the system's influencing factors and degrees of freedom in the actual research in order to construct a reasonable lumped-parameter model.

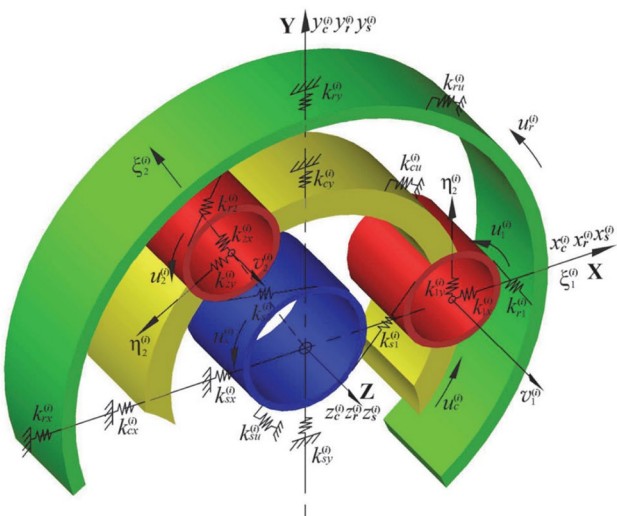

**Figure 3.** Lumped-parameter model of single-stage planetary gear system [20].

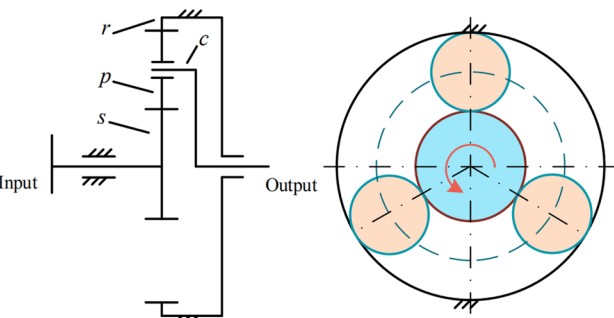

**Figure 4.** 2K-H planetary gear set [23].

### 2.2. Finite Element Model

In order to overcome the defects of the lumped-parameter method, scholars have carried out in-depth research on the finite element model. The finite element model is based on both the overall assembly model of the system and the finite element method to divide the system into a finite number of nodes and units, and it is obtained by defining the interaction relationship between the components, nodes, and units [27].

Parker et al. [28] established the finite element model of the planetary gear mechanism for the first time. The research shows that the system's natural frequencies and mode shapes obtained via the finite element model and the lumped-parameter model are consistent. In 2003, Kahraman et al. [29] established a fully flexible finite element model of a spur planetary gear train and studied the influence of gear compliance on dynamic stress and wheel rim deflection. In 2005, Singh [30] established a three-dimensional finite element model of a helical planetary gear train based on the two-dimensional finite element method. The model was used to study the relationship between the tangential error of gear pinhole and the number of planetary wheels and load distribution coefficient. In 2009, Abousleiman et al. [31] proposed a finite element model that can simulate the dynamic behavior of spur gear and helical gear planetary transmissions. In 2017, based on the ADAMS software, Salagianni et al. [32] considered the influence of gear backlash and friction in the mixed lubrication process and used the combined friction dynamics modeling method to conduct a modal analysis of the planetary gear system. In 2018, Liu et al. [33] established the dynamic finite element model of the planetary gear train. The friction force in the planetary gear set was calculated using the Coulomb friction model. The effects of rotational speed, torque, and fault width on the dynamic characteristics of planetary gear trains were discussed. In 2019, Zhang et al. [34] established a transient dynamic analysis model of a two-stage planetary gear train using the finite element method based on the schematic diagram of the two-stage planetary gear transmission shown in Figure 5.

In 2021, Sang et al. [35] established a three-dimensional model of the 3K-II planetary gear train, as shown in Figure 6. The time-varying stiffness of the internal/external mesh pair under different gear tooth root crack faults was calculated by using the finite element method. Figure 7 is the finite element model of the external (a)/internal (b) meshing gear pair. On this basis, the time domain and frequency domain characteristics of the lateral vibration signal and output torsion signal are studied.

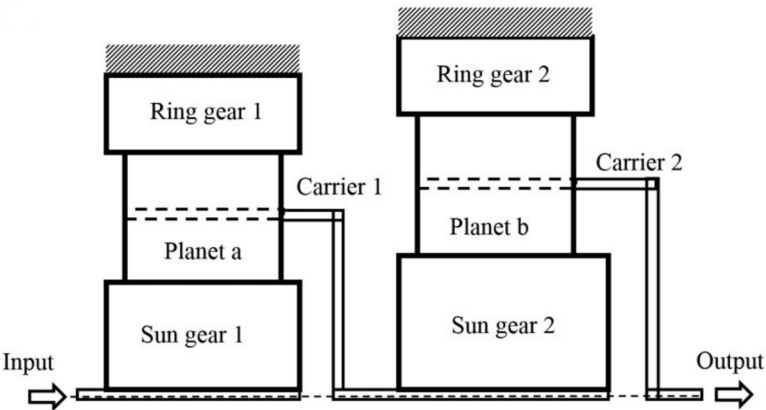

**Figure 5.** Schematic diagram of two-stage planetary gear transmission [34].

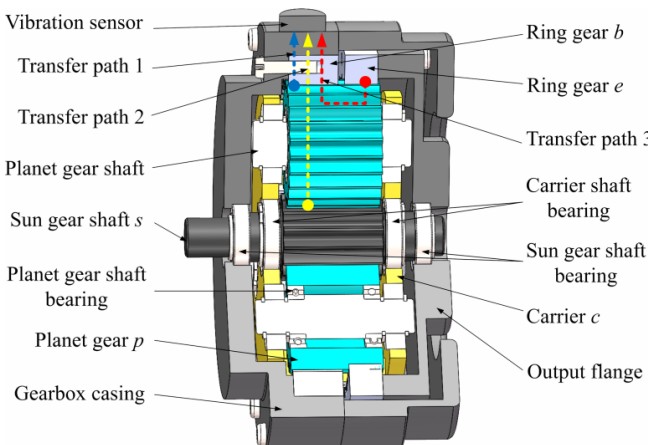

**Figure 6.** Three-dimensional model of 3K-II planetary gear set [35].

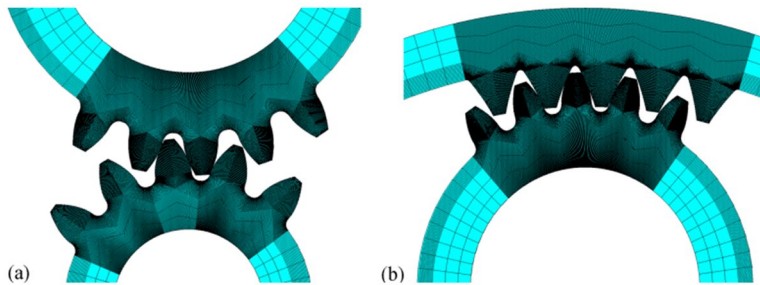

**Figure 7.** Finite element model of external (**a**)/internal (**b**) gear pair [35].

In 2021, Wang et al. [36] used the finite element method to divide the system into shaft, meshing, and bearing units. A full-degree-of-freedom dynamic model of the herringbone planetary gear transmission system was established considering the time-varying mesh stiffness and comprehensive mesh error. Equation (1) is the derived overall differential equation of motion. It is found that the flexible deformation of the shaft has a significant

impact on the dynamic characteristics of the system, which was calculated using the following equation:

$$M_z \ddot{Y}(t) + C_z \dot{Y}(t) + K_z Y(t) = F_z \qquad (1)$$

where $M_z$ is the overall stiffness matrix of the system; $C_z$ and $K_z$ are the overall damping and stiffness matrixes of the system; $F_z$ is the external excitation force array; and $Y(t)$ is the displacement column vector of all the nodes in the system.

According to the above research, it can be found that the establishment of the finite element model can obtain a higher calculation accuracy, consider the influencing factors more comprehensively, and better reflect the actual engineering problems. However, when the system structure is too complex, the finite element model often has problems, such as a complex pre-processing process, a large number of degrees of freedom, difficult calculations, and a long calculation time. With the development of computer technology, this method is becoming a powerful tool for studying the dynamic characteristics of planetary gear systems.

### 2.3. Rigid—Flexible Coupling Model

It is not difficult to see that the above two models have their shortcomings. That is, the fidelity of the lumped-parameter model is insufficient, and the finite element model has the disadvantage of a large amount of calculation. Therefore, scholars have proposed a rigid–flexible coupling model, which is also known as a finite element/semi-analytical model. It is an analysis model that regards some components in the system as rigid bodies and some components as flexible bodies. At present, the internal ring gear, or the thin-walled spoke gear, is mainly treated flexibly while other components are still considered rigid bodies, and the contact relationship between the tooth surfaces is calculated using an analytical method.

In 2001, Kahraman et al. [37] used the finite element/semi-analytical non-linear contact mechanics method to model the typical planetary gear mechanism of an automotive automatic transmission. In 2006, based on the modal condensation method, Velex [38] established a hybrid three-dimensional finite element/lumped-parameter model and used it to analyze the planetary gear dynamics of the flexible internal ring gear. In 2009, Ma et al. [39] established the rigid–flexible coupling model of the planetary gear and studied the dynamic response characteristics of the planetary gear train under extreme conditions. In 2012, Parker et al. [40] studied the non-linear dynamics of planetary gears with elastic ring gears and gave frequency response functions for primary, subharmonic, and superharmonic resonances. In 2021, Wang et al. [41] analyzed the influence of faulty gears on the dynamic characteristics of planetary gear systems and conducted a joint simulation analysis using rigid–flexible coupling dynamic models. In 2020, Xu et al. [42] considered the coordination conditions of torsional compliance and axial sliding of the left and right teeth of the sun gear and established a dynamic model of the herringbone planetary gear system based on the parameter concentration method, as shown in Figure 8. Studies have shown that with increasing torsional compliance (decreasing torsional stiffness), the mesh stiffness and maximum tooth surface load of the sun and planet gears both increase on the left (input side) and decrease on the right. In the figure, O-XYZ is the static coordinate system. The *X*-axis goes through the first planet's rotation center and uses the rotation center of the carrier as the origin, which rotates at a constant speed around the *Z*-axis with the angular velocity $\omega_c$ of the carrier.

Scholars have proven the feasibility of the rigid–flexible coupling model through research. It has the advantages of a fast solution speed, low requirements for computer performance, and a high calculation accuracy—and, to a certain extent, it combines the advantages of the lumped-parameter model with the finite element model. However, it is essentially a complex multibody dynamic analysis model and the coupling effects of various parameters need to be considered in the modeling process, which is still more complicated than the concentrated parameter model.

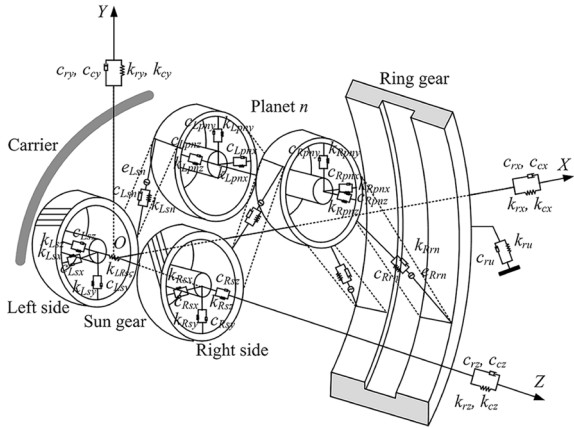

**Figure 8.** Dynamic model of herringbone planetary gear system [42].

*2.4. Comparison of the Three Models*

Through the above research, it is found that the three models have their own advantages and disadvantages. Table 1 gives the advantages, disadvantages, and main uses of the three models. When building a model, it is necessary to combine the actual working conditions and the system characteristics, select an appropriate modeling method, and obtain the solution results that meet the requirements. The centralized parameter model has a simple modeling process and a fast solution speed and is comparable to the finite element model and the rigid–flexible hybrid model in terms of accuracy in analyzing the dynamic response of the system. Therefore, the dynamic analysis of the multi-stage planetary gear transmission system using the lumped-parameter model is accurate and efficient. This is why lumped-parameter models are widely used.

**Table 1.** Comparison of the characteristics of the three models.

| Model | Advantages | Disadvantages | Application |
|---|---|---|---|
| Lumped-parameter model | Simple structure; Rapid modeling; Convenient solution | Less freedom; The solution accuracy is low; It is difficult to simulate the actual engineering. | A reasonably simplified dynamic model can be quickly constructed according to specific problems. |
| Finite element model | High calculation accuracy; More consideration of factors | The pre-processing and post-processing processes are complex, and the solution speed is slow. | It can provide an accurate system dynamic analysis model and can be used to verify the centralized parameter model. |
| Rigid–flexible coupling model | Solution speed is fast; Computer performance is not high; The calculation accuracy is high | The coupling problem of various parameters needs to be considered comprehensively, and the model is more complicated. | Compared with the lumped-mass model, it is closer to the real working conditions of the system, and the calculation speed is faster than that of the finite element model, which has broad application prospects. |

In addition, the centralized parameter model and the finite element model have their own characteristics and the two can be used in combination. In 2007, Ambarisha et al. [12] established the finite element and lumped-parameter models of the planetary gear transmission system and found that the results of the two models were consistent. In 2013, Ericson et al. [43] used the finite element method to establish a planar finite element contact model of the planetary gear system. The system's dynamic characteristics were explored using the finite element method, lumped-parameter method, and experimental modal analysis method, respectively. The experimental modal analysis method proves the validity of the finite element method and the lumped-parameter method. In 2015, He et al. [44] established the gear–rotor dynamic model of the planetary gear transmission system with the lumped-parameter method and established the box dynamic model with the finite element method. The two were coupled

through the deformation coordination condition, and the system's dynamic behavior before and after the coupled box was studied.

In order to overcome the shortcomings of the low precision of the lumped-mass method and the difficult calculation of the finite element method, scholars continue to propose new modeling methods. In 2011, Wasfy et al. [45] proposed a high-fidelity multibody dynamic model for predicting the transient response of planetary gear trains. The model takes into account the effects of gear mesh stiffness, damping, friction, and backlash. In 2017, Wei et al. [46] proposed a coupling dynamic modeling method for planetary gear transmission systems considering the structural flexibility based on finite nodal elements. This method has a higher accuracy than the lumped-mass method and a faster calculation speed than the finite element method. In 2018, Hou et al. [47] adopted the nodal finite element method to establish a dynamic model of the herringbone planetary gear system considering friction, mesh stiffness, and mesh error excitation. The influence of friction excitation on the dynamic response of the herringbone planetary gear under different working conditions was studied. In 2023, Liu et al. [48] calculated the nonlinear dynamic model of a spur gear system, considering the dynamic increment and velocity-dependent grid stiffness effect based on the analytical finite element method, and obtained a dynamic response.

In addition, it can be found that the core content of the lumped-parameter model is to simplify the component structure into particles and build the model by establishing the connection between the particles. The core content of the finite element model is to discretize the structure into nodes and units and build a dynamic model by adding boundary conditions. The rigid–flexible coupling model is to convert the parts in the system that need to consider the structural deformation characteristics into flexible bodies, and the rest of the components are still regarded as rigid bodies. When considering the non-inertial system, the planetary gear transmission system will be affected by additional loads such as drag inertial force, Coriolis inertial force, and gyro moment [49]. That is, the main influence is the coordinate system's setting of the system, the additional excitation effect, the coupling relationship, etc. Therefore, the three traditional research models are still applicable to the study of the dynamic characteristics of the planetary gear system in the non-inertial system.

## 3. Research on the Transmission Systems under the Non-Inertial System

The so-called non-inertial system is the frame of reference for variable speed movement relative to the inertial system [50]. The mechanical conditions of the system will change in the non-inertial system. For the transmission system, the non-inertial environment generally includes the maneuvering flight of the aero engine with the random body and the gearbox's movement with the vehicle's speed change, etc. At present, the research on the vibration characteristics of the transmission system under the non-inertial system mainly focuses on the gear–rotor system. Scholars have found that the vibration characteristics of the rotor system are different under the maneuvering flight and the static ground state [51–54]. However, there are few studies on the non-inertial dynamics of gear transmission systems. Planetary gear systems have been widely used in non-inertial environments such as aviation, aerospace, automobiles, and ships. Therefore, this section briefly introduces the non-inertial dynamics and summarizes the research status of the non-inertial dynamics of gear–rotor systems and planetary gear systems.

### 3.1. Non-Inertial Dynamics

When studying the dynamics of a non-inertial system, it is first necessary to set up a non-inertial reference system, analyze the kinematics of the research object, and find the relationship between the fixed coordinate system and the moving coordinate system [50]. Then, the dynamic equation of the system is constructed by introducing Newton's second theorem of inertial force, the Lagrange equation, and the Hamilton principle in relative form. Additional accelerations, drag inertial forces, Coriolis inertial forces, energy issues, etc., need to be considered when building the model. At present, the research on non-

inertial dynamics mainly focuses on systems such as rotors, rotating machines, planar mechanisms, and flexible beams.

In 1983, Sony et al. [55] conducted a dynamic study of a rotor mechanical system under fundamental excitations. The effects of both the gyro and Coriolis effects on the system response were studied. In 1987, Simo et al. [56] studied the application of the non-linear theory in the transient dynamic analysis of flexible structures. The study found that, for rotating structures, the use of geometrically non-linear (at least, second-order) beam theory needs to be considered when studying the effect of centrifugal forces on bending stiffness. For rotating planar beams, differential equations of motion for coupled tensile and bending deformations that account for all inertial effects (Coriolis forces, centrifugal forces, and rotational accelerations) can be derived from the fully non-linear beam theory. In 2007, Yang et al. [57] studied the influence of the aircraft's dive and pull-up motion on the dynamic characteristics of the rotor system for the Jeffcott rotor system with cracks. In 2014, Ni et al. [58] proposed a description method for both a helicopter space maneuvering flight and a tail drive shaft motion pose. In 1999, Wang et al. [59] proposed a bond graph method for the dynamics of planar mechanism systems in non-inertial systems and derived a unified formula for the system state equation. In 2001, Han et al. [60] established the relative dynamic equation of the blade system in a non-inertial reference frame by using the Lagrange equation and the finite element method. In 2012, He et al. [61] studied the dynamics of large-scale motion and non-linear deformation space flexible beams in a non-inertial reference frame. In 2015, Liu et al. [62] proposed a non-linear accurate planar flexible beam structure model. The finite element method was used to discretize the beam structure, and the Lagrange method was used to establish the precise dynamic equation of the system in the non-inertial system. In 2016, Li et al. [63] established a rigid–flexible coupling system equation for a flexible beam with dynamic stiffness by using the mechanic's theory and subsystem modeling principles in a non-inertial system.

It can be found that problems concerning non-inertial system dynamics widely exist in various working environments. With the increasing requirements for the authenticity and accuracy of the study of dynamic characteristics, the study of non-inertial dynamic behavior will become one of the hotspots.

*3.2. Gear–Rotor System*

Scholars have found that the gear–rotor system will be subject to additional excitation caused by the basic motion of the system in practical work. Taking the aircraft rotor system as an example, the basic motion includes level flight, dive, hover, climb, various difficult flights, etc. Wen et al. [64] think that the influence of aircraft maneuvering flight on the dynamic characteristics of rotor systems can be regarded as a kind of basic excitation. In 2006, Duchemin et al. [65] derived the motion equation of the rotor system under the basic motion condition by using the Lagrange energy method. In 2010, El-Saeidy et al. [66] derived the dynamic equation of the rigid rotor using the six degrees of freedom under the coupling action of the foundation motion excitation and the mass unbalance mechanism. In 2014, Hou et al. [67] studied the dynamic response of the friction-impact rotor system during hovering and analyzed the influence of non-inertial environments on the non-linear dynamic behavior of the rotor system. In 2016, Saxena et al. [68] used the ANSYS finite element analysis software to conduct a modal analysis of the ball-bearing gear–rotor system. The influence of the bearing stiffness variation on the gear–rotor system's natural frequency was studied. Figure 9 is a schematic diagram of the gear– rotor system; the unit is mm. In 2018, Han et al. [69] used the finite element method to establish a dynamic model of the rotor bearing system and studied the irregular transient vibration of the system during maneuvering flight. In 2019, Soni et al. [70] studied the vibration response of the maglev rotor system when the airframe pitched, rolled, and yawed. In 2021, Han et al. [71] established a multi-stage gear–rotor transmission system model using the finite element method. The steady-state response of the system caused by the unbalanced quantities and static transmission errors was studied.

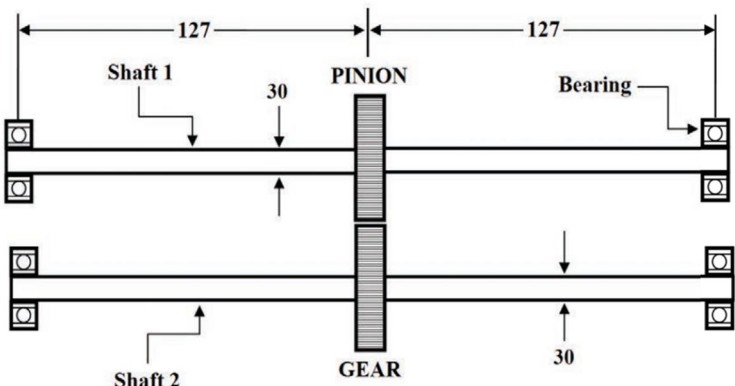

**Figure 9.** Schematic diagram of gear–rotor system [68].

According to the above research, it can be seen that there has been much research on the non-inertial dynamic characteristics of the gear–rotor system. Scholars have studied the dynamics of various non-inertial flight environments and the research is still deepening. Considering the similarity between the research on the dynamics of the gear–rotor system and the planetary gear system, scholars can advance the research on the dynamics of the planetary gear system in the non-inertial system based on the research method of the rotor system in the non-inertial system.

### 3.3. Planetary Gear System

Planetary gear transmission is widely used in many fields due to its excellent transmission performance, including in non-inertial motion environments. However, the research on the non-inertial dynamics of the gear transmission system has only attracted the attention of scholars in recent years. Although the non-inertial dynamics of the rotor system provide research references, the gear transmission system is a strongly non-linear system with self-excitation characteristics, and its dynamic behavior is significantly different from that of the engine rotor system [4].

In 2013, Chu et al. [72] studied the dynamic behavior of a gear–rotor system under periodic fundamental motions (rolling, pitching, and yawing motions). The results of the study show that no matter what type of base motion occurs, there is an increase in the transient response of the gear–rotor system. In 2015, Liang et al. [73] established a translational–torsional lumped-mass model for a spur planetary gear train. The vibration characteristics of the system under the actions of gyro torque and centrifugal force were studied. In 2018, Liu et al. [74] established a gear-sliding bearing coupling dynamic model under the action of ship swaying. The non-linear oil film force was introduced into the model, and the meshing force of the tooth surface and the back side of the tooth was also considered. The vibration response of the sliding bearing-parallel shaft gear pair when the foundation is swaying was studied. In 2018, Qiu [75] elaborated on the modeling method of the basic motion dynamic model of planetary gears. The translation–torsion axial dynamic model of the planetary gear train under the basic pitch motion was derived. In 2019, Wei et al. [49] considered the comprehensive influence of the internal non-inertial system and the external non-inertial system of the planetary gear transmission system under the space environment non-inertial system and studied the law of motion change as applicable to the system. In 2021, Wei et al. [76] established a dynamic model of the planetary gear transmission system under a time-varying pose. In 2023, Peng et al. [77] established a dynamic planetary gear transmission system model under the non-inertial system. The study defined the stress non-inertial system coefficients in order to describe the influence of the additional effects of the non-inertial system on the dynamic stress of the planetary gear train. The research results show that the random body maneuverability of the stress non-inertial coefficients increases in a non-linear manner as a whole. The influence of non-inertial system conditions cannot be ignored under high maneuvering conditions.

Figure 10 is a schematic diagram of the non-inertial coordinate system of the planetary gear system. All the degrees of freedom of the sun gear shaft and the ring gear, as well as the translational degrees of freedom of the planet carrier shaft, are defined in the planet carrier moving coordinate system, $o_c$-$x_cy_cz_c$. All the degrees of freedom of the planetary gears and pins are defined in the planet carrier follow-up coordinate system (the origin is at the center of the planetary gear), $o_p$-$\xi\eta z_p$. The torsional degrees of freedom of the planet carrier and all the degrees of freedom of the shell are in the coordinate system, $o_0$-$x_0y_0z_0$.

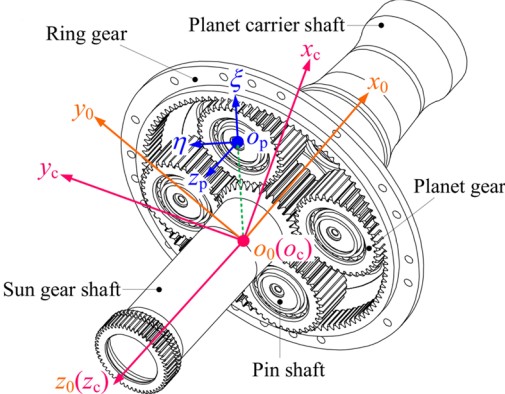

**Figure 10.** Planetary gear system non-inertial coordinate system setting [77].

The above research shows that the current research on the non-inertial dynamics of the planetary gear system is still in its infancy as there are only a few research scholars. Among them, Professor Wei Jing from Chongqing University has conducted in-depth research on the construction of the dynamic model of the planetary gear system in the non-inertial system, as well as analysis methods and the characteristics of the impact of additional excitation. Relevant researchers can refer to this section to rationally combine the motion analysis method, the additional item analysis method, and the basic motion characteristics of the non-inertial rotor system dynamics with the research on the dynamics of the gear system and continue to broaden the research on the dynamics of gear transmission systems in non-inertial systems.

## 4. Dynamic Characteristics

Considering various influencing factors to establish a dynamic model for simulating the actual working conditions of the system, mastering the influence of various factors on the dynamic characteristics is the key to developing a planetary transmission system with better transmission performance. At present, scholars have conducted in-depth research on the dynamic characteristics of the planetary gear system under the inertial system. The main content includes the inherent characteristics of the planetary gear system, vibration characteristics, dynamic load characteristics, contact characteristics, dynamic load stability, etc. It requires a research method that will not deviate from the traditional dynamic characteristics when researching the non-inertial dynamics of planetary gears. Therefore, this section mainly focuses on the research status of dynamic load characteristics, vibration characteristics, and vibration control. It is expected to provide a reference for the study of the dynamic characteristics of the planetary gear system in the non-inertial system.

### 4.1. Dynamic Load and Load-Sharing Characteristics

The system load of the planetary gear system changes dynamically due to the influence of factors such as external excitation, manufacturing and installation errors, backlash, time-varying mesh stiffness, mesh damping, meshing impact, etc. Under the conditions of high speed and high load, if the distribution of the dynamic load is uneven, it will cause problems, such as an unbalanced load and excessive dynamic stress, which will reduce the fatigue strength of the gear teeth, system reliability, and service life. Therefore, scholars have

carried out quite a lot of research to improve the dynamic load stability and load-sharing characteristics of the system and to ensure the stable and safe operation of the system.

As early as 1980, Hidaka et al. [78] established a purely torsional dynamic model of a planetary gear train. The dynamic load on the tooth surface of the planetary gear was studied. In 1984, Ma et al. [79] established a planetary gear dynamic model considering the influence of time-varying mesh stiffness, flexible ring gear, tooth profile error, and other factors. The problem of dynamic loads and load distribution occurring in the planetary gears of a PT6 turboprop gearbox was investigated. In 1994, Kahraman [80] proposed the static load sharing factor, dynamic load sharing factor, and dynamic load factor to describe the load-sharing characteristics of the 2K-H planetary transmission system. In 2005, Bodas et al. [81] used experimental methods to analyze the influence of the manufacturing error of the planet carrier and gear on the static load-sharing characteristics of the planetary transmission system. In 2017, Dong et al. [82] established a non-linear dynamic model of the planetary gear transmission system considering manufacturing error, assembly error, and gear flexibility and studied the load-sharing characteristics of the system. Figure 11a is the coordinate definition diagram of the planetary gear system and Figure 11b is the dynamic model.

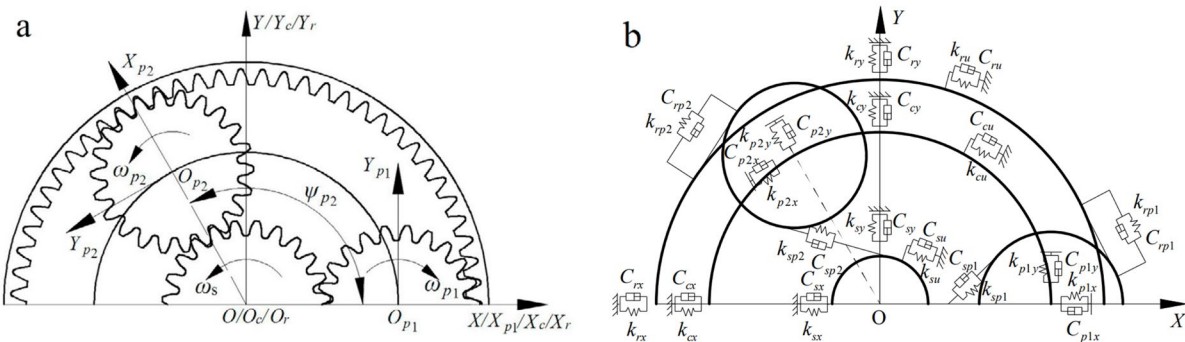

**Figure 11.** Planetary gear system non-inertial coordinate system setting [82].

In Figure 11a, *OXY* is the global coordinate system. $O_C X_C Y_C$ and $O_r X_r Y_r$ are the inertial coordinate system of the planet carrier and the sun gear. $O_{pi} X_{pi} Y_{pi}$ are respectively the follow-up coordinate system of the *i*th planetary gear (*i* = 1,2,3). In the Figure 11b, the carrier is coupled with the planet pin-bearing assembly via stiffness $k_{cp}$ and damping $C_{cp}$. The carrier support bearing is modeled by stiffness $k_c$ and damping $C_c$. The sun gear connects to planet gears along the line of action through gear mesh stiffness $k_{jp}$ and damping $C_{jp}$. The sun and ring gears coupled to the ground or shaft via support stiffness $k_j$ and damping $C_j$, also a torsional stiffness $k_{ju}$ and damping $C_{ju}$, (*j* = s,r).

Leque et al. [83] established a single-stage planetary gear train 6N-DOF (N is the number of components) dynamic model. The load-sharing characteristics of the system were studied. Iglesias et al. [84] studied the influence of eccentric errors and planetary pin axis errors on the load sharing of the planetary gear train under quasi-static conditions. In 2018, Zhang et al. [85] used the translation–torsion lumped-parameter model to study the influence of multiple meshing phases on the load sharing of herringbone planetary gears. Hu [86] proposed a load distribution model of the helical planetary gear train, which can predict the influence of support stiffness, pin position error, runout error, tooth thickness error, meshing phase, and tooth surface modification on the tooth surface load. In 2019, Hu et al. [87] improved the definition of dynamic load and load sharing coefficients of the planetary gear train. In 2020, Chung et al. [88] found, through simulation, that floating components can effectively improve the problem of uneven load distribution caused by non-torque loads and pinhole position errors. In 2022, Peng et al. [4] established the dynamic model of the planetary gear transmission system based on the idea of node finite element and by considering the time-varying mesh stiffness, tooth backlash, tooth profile error, internal inertia system, and external inertia system. The influence of maneuvering

space motion on the load distribution of planetary gear transmission was studied. The system-level coupling dynamic model is shown in Figure 12. From the above research, it can be found that the main purpose of the current research on uniform load and dynamic load characteristics was to explore the influence characteristics of various objective factors inside and outside of the planetary gear system, such as dynamic excitation, error, inertia load, and so on. However, most studies ignore factors such as the non-inertial load, system flexibility, etc. In addition, it can be found that the finite element model is better than the lumped-parameter model. Combined with experimental research, the load characteristics of the system can be better explored.

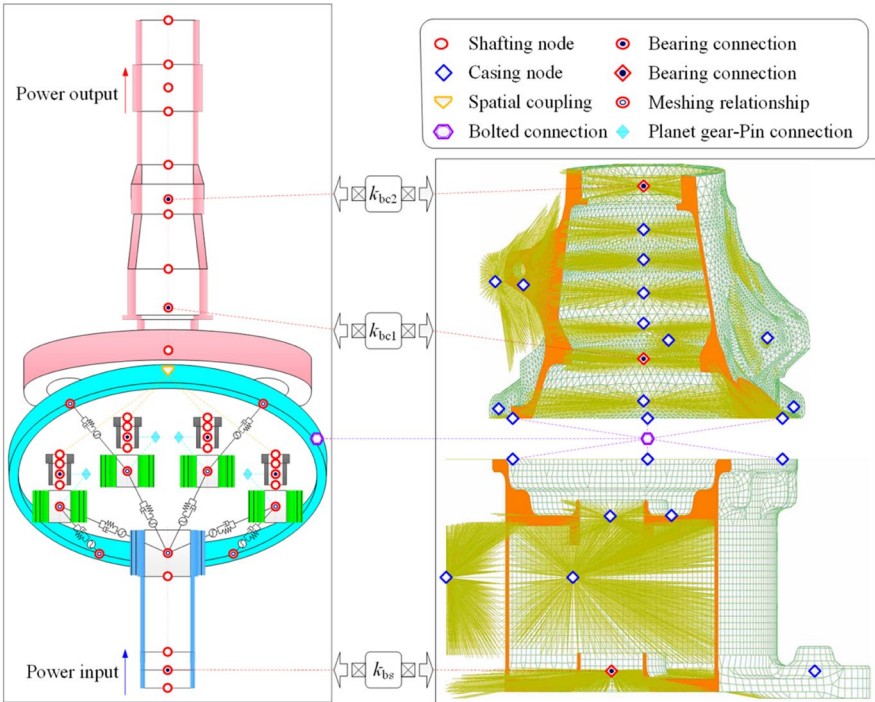

**Figure 12.** System-level coupling dynamic model of planetary gear transmission [4].

### *4.2. Vibration Characteristics*

Vibration theory is the basis of the study of planetary gear dynamics. Vibration and noise problems have always been the main content of gear system dynamics. Vibration response mainly includes displacement, velocity, acceleration, and so on. By studying the vibration response of the system under the action of various factors, the vibration and stress law of each component of the system can be obtained, which provides guidance for the structural design, vibration and noise control, fault diagnosis, and so on. A relatively perfect theory and experimental research method of planetary gear system vibration characteristics have been formed through scholars' research.

Ferraris et al. [89] studied the vibration characteristics of the double-rotor system with casing using the finite element method and verified the simulation results through experiments. In 2000, Edwards [90] balanced the rotor–support–foundation system and identified the mass, stiffness, and damping parameters of the flexible foundation through experiments. In 2008, Stringer [91] established a six-degrees-of-freedom gear–rotor system model for a helicopter main reducer. At the same time, considering the gyro matrix, the vibration response of the helicopter main reducer was analyzed. From 2008 to 2010, Inalpolat et al. [92,93] analyzed the influence of time-varying mesh stiffness, gear manufacturing error, and input speed on the dynamic response amplitude of multi-stage planetary gear systems of the automatic gearbox. In 2016, Zhang et al. [94] established the corresponding dynamic models of the two-stage planetary gear transmission system by using both the lumped-mass method and the finite element method to compare and analyze their dynamic

characteristics. In 2016, Srikanth et al. [95] carried out a dynamic analysis of the wind turbine transmission system under random aerodynamic loads. In 2020, Xiao et al. [96] established the dynamic model of the planetary gear transmission system by using the lumped-parameter method, and the dynamic responses of normal and fault systems were obtained. In 2021, Min et al. [97] established the non-linear dynamic model of planetary spur gear transmission systems using the mass concentration method. The influence of tooth clearance on the gear vibration response is obtained via the phase plane diagram and the FFT spectrum diagram. In 2020, Tatar et al. [98] used a three-dimensional dynamic model to study the influence of the parameters on the dynamic response of planetary gear rotor systems, including the number of planetary gears, planetary gear mass, gear meshing stiffness, planetary gear speed, etc. Figure 13 shows a schematic diagram of the planetary gear–rotor system.

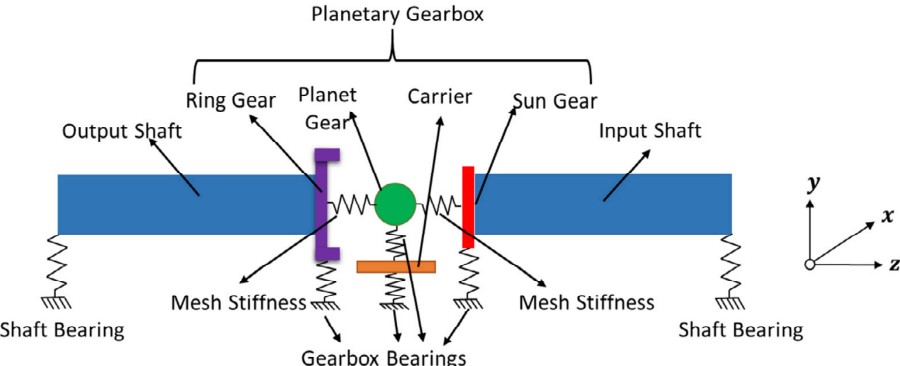

**Figure 13.** The planetary gear–rotor transmission system [98].

It can be seen that vibration problems exist widely in all kinds of transmission systems. In addition, scholars have begun to study the vibration problems of gear faults and gearbox coupling systems. The factors considered are becoming gradually more comprehensive and new analysis methods are constantly emerging.

*4.3. Research on Vibration Control*

As a result of the planetary gear transmission system structure being more complex, the vibration problem of the planetary gear transmission system is more obvious than that of the fixed shaft gear system. At present, the ideas of vibration control can be summarized as vibration reduction and vibration isolation. Scholars have studied from the point of view of the gear parameters and structure optimization, so as to increase system damping, feedback control, and so on. The vibration and noise of the system are controlled by improving the accuracy of the gear manufacturing and installation, modifying the tooth root and tooth top, adding dampers, isolating vibration sources, and so on.

In 1986, Tavakoli et al. [99] first proposed a discussion on modifying involute gears, which modifies the tooth tops of large and small gears according to the length of the double-tooth meshing section on the meshing line. In 2013, Bahk [100] et al. analyzed the influence of profile modification on the static and dynamic characteristics of gear systems based on this model. In 2016, Hui et al. [101] obtained the best tooth profile modification curve aimed at static transmission errors and gear vibration amplitude response. The relationship among the tooth profile modification curve, the tooth profile modification amount, and the gear modification coefficient was studied. In 2019, Sánchez et al. [102] determined the modification parameters of the gear by meshing stiffness, load distribution, and transmission error. In 1975, Okamura et al. [103] studied the vibration and noise of the gear system by adding a damping ring to the spur gear. It was found that the damping ring has a significant effect on the noise and vibration frequency of the gear transmission. At the same time, the influence of different damping ring parameters and installation errors on the gear system vibration and noise was also studied. In 2020, Lu et al. [104] studied the influence of structural parameters of the integral squeeze film damper on the radial

stiffness of the damper and established a vibration control method of the gear system based on the damper. The effects of the installation position and damping fluid on the damping characteristics of gear shafting were studied. In 2019, Olanipekun et al. [105] applied the pole placement theory to the planetary gear system to avoid resonance. The control strategy was to provide feedback on the displacement and speed response of the system output. The natural frequency of the planetary gear system was changed by the displacement feedback. Active damping was added using velocity feedback to avoid resonance of the system. Figure 14 is a schematic diagram of a closed-loop feedback system, where a disturbance is applied to the input and the output is fed back to achieve system stability.

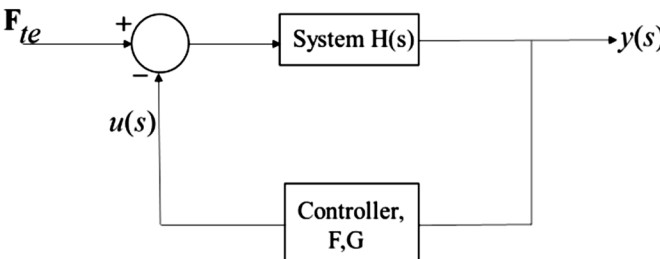

**Figure 14.** Schematic diagram of closed-loop feedback control system [105].

In short, the current vibration control research can be divided into active control and passive control. The fundamental purpose of vibration and noise control is to study the influence of the different methods on the vibration characteristics of the system and to avoid the natural frequency to avoid resonance. Suppressing the system vibration by optimizing gear manufacturing precision and structure has fallen into a bottleneck. The active control of vibration and noise based on the control principle will be a research hotspot in the future.

## 5. Conclusions and Outlook

The planetary gear system is widely used because of its excellent transmission performance. Its structural form and working environment are relatively complex, which will inevitably produce dynamic load, vibration, noise, and other phenomena during work. In addition, some planetary gear systems work in non-inertial environments such as those with high maneuverability, which changes their dynamic characteristics. In order to promote the study of the dynamic characteristics of planetary gear systems under the non-inertial system, this paper summarizes the lumped-parameter model, the finite element model, and the rigid–flexible coupling model used in the study of traditional planetary gear dynamic characteristics. The advantages, disadvantages, and application principles of the three analysis models are found and the two models can still be used when considering non-inertial factors. Then, the basic theory of non-inertial dynamics is briefly described. The research status of the dynamic characteristics of non-inertial gear–rotor systems and planetary gear systems are summarized. It is found that there is something in common between the two studies and the motion analysis method; additional term analysis methods and basic motion characteristics of non-inertial rotor system dynamics can be reasonably combined with the study of gear transmission system dynamics. Then, the dynamic load and load-sharing characteristics, vibration characteristics, and vibration control of the planetary gear system are summarized. Finally, the future research direction is prospected to provide reference for relevant scholars and further improve the planetary gear system's life, performance, and reliability.

The following are suggestions for the future direction of development:

1. The non-inertial dynamics of the planetary gear system should be studied. At present, the research on the non-inertial dynamics of transmission systems is mainly focused on rotor systems. The traditional research method is no longer applicable to the planetary gear system in the non-inertial system. There is still a lot of room for development in the study of non-inertial dynamic characteristics of planetary gear systems.

2. The vibration control of planetary gear systems needs to be studied in depth. The vibration problem is the core problem in the study of gear transmission dynamics. The development of more efficient vibration and noise control methods is the basis for improving transmission performance.

3. The dynamic characteristics of multi-stage planetary gear systems need to be studied. The multi-stage planetary gear transmission structure is more complex, more factors need to be considered in the study of dynamic characteristics, and the dynamic model is more complex. The analysis of its vibration characteristics is more difficult. So, it is particularly important to study the dynamic characteristics of the compound planetary gear system.

4. According to the actual working situation of the system, it may be necessary to explore the influence law of more factors on the dynamic characteristics of the system. The research on modeling techniques, solving techniques, analysis, and verification methods for dynamic characteristics of the planetary gear system is constantly enriched.

## 6. Patents

None.

**Author Contributions:** B.G. provided research topics to guide the research contents of this article. Y.W. sorted out existing research materials and prepared the manuscript. G.Y. provided content suggestions and revised the manuscript. All authors have read and agreed to the published version of the manuscript.

**Funding:** This work was supported by the National Natural Science Foundation of China (Grant No. 52075134), the Major Science and Technology Project in Heilongjiang Province: key Technology Research and demonstration Application of Aeronautical Accessories Transmission system (Grant No.2021ZX04A01), and the Key R & D Project in Heilongjiang Province: cooperative Research on key Technologies of deformation Control in Aeronautical thin Web Gear Machining (Grant No.2022ZX07D02).

**Institutional Review Board Statement:** Not applicable.

**Informed Consent Statement:** Not applicable.

**Data Availability Statement:** Not applicable.

**Conflicts of Interest:** The authors declare no conflict of interest, financial or otherwise.

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
