# Peer review of "Research Progress on the Dynamic Characteristics of Planetary Gear Transmission in a Non-Inertial System"

_machines, doi:10.3390/machines11070751_

Round 1

Reviewer 1 Report

This review describes recent developments in planetary gear systems in non-inertial frames. The contents are very novel. This manuscript could provide new research ideas for other researchers. However, there are still some shortcomings that need to be improved. Here are some suggestions that should be considered in the revision.

1. The language needs to be improved. Check the manuscript carefully for spelling and grammatical errors.

2. Provide necessary explanations for the symbols in Figures. Such as Figure 4 and Figure 8.

3. Simplify the introduction of some cases in the manuscript.

4. The target and contributions of the research should be specified at the end of the introduction.

I believe that the submitted manuscript can be accepted after correcting. I also believe that this manuscript may be helpful to scholars working in this field.

Minor editing of English language required

Author Response

We have revised the manuscript according to your comments. Please refer to the attachment for our reply to the comments on the manuscript review.

Reviewer 2 Report

This paper summarized the methods and results of dynamic analyses of planetary gear system. But there are several obvious weaknesses of this paper:

1. Although the dynamic modeling methods and analyses of planetary gear systems are discussed in detail, there are not enough text for the studies on planetary gear systems in non-inertial system. The main reason is that there are still few studies in this area, indicating that its impact may not be so obvious in many occasions to prompte enough scholars to have interests on it and form a referable review. It is not appropriate to write a review with this narrow topic.

2. In addition to the lumped mass method and finite element method mentioned in the paper, there are actually some rigid-flex coupling models, such as the model considering the flexible ring gear, but the author did not discuss them.

3. The study of rotor systems in section 3.2 is of little relevance to this paper, and the studies of the dynamics of non-inertial systems of planetary gear system are not mentioned in section 4.

Therefore, the structure of this paper needs to be reorganized, and useful literature should be added and irrelevant literature should be deleted.

  • English expression meets the requirements of scientific papers.

Author Response

(The authors gave the same response as above.)

Reviewer 3 Report

The authors have produced a literature review that ranges on different aspects of planetary gear dynamics. However, there are few points that needs to be addressed before this manuscript is ready for publications.

Major:

1.     The authors have put a lot of effort in describing the equation of motion with the purpose to describe NVH characteristic of planetary gear set and yet the is very little to no mention of the works that went into calculating friction.

a.     One can argue that the method of calculating friction can have significant implications of the final dynamic response.

b.     It is suggested that the literature review should have a dedicated section of friction alone, further details are described in the following point.

2.     Its is clear that friction is directly related to the material the gear teeth are made from, depending on whether lubrication is needed or not.

a.     What is not clear in the abstract is that only gear sets found in aviation industry is being considered. Is this the authors intent?

                                               i.     If so, then there should be a dedicated section on the research of lubrication theory, or the minimum of how lubricant is generally modelled to determine friction.

                                              ii.     If not, then discussion on how friction is modelled generally should be added. Example the difference in plastic gears that have little to no added lubrication compared to traditional metal gear teeth which does.

3.     The Reviewer is struggling to justify the current strength of the manuscript’s’ novelty and content to warrant publication.

a.     Can the authors produce a clear and concise statement that illustrates how this literature review is comprehensive enough for publication. This will help support the content.

b.     The abstract is broken down well but a little too long and does not provide a (concise) statement that allows the reader to understand the major contribution of this manuscript

Minor

·       Please remove the heading in the abstract, this is not necessary (‘Background, ‘Method’, ‘Results’ and ‘Conclusion’

o   Abstract should not have headings and should be a bit more concise.

Please check abstract again, sentences seem to end abruptly or do not follow well to the next point. 

Author Response

(The authors gave the same response as above.)

Round 2

Reviewer 2 Report

The revised version has been improved a lot according to the comments and can be accepted.

Reviewer 3 Report

The Reviewer accepts the amendments by the Authors. The manuscript is ready for publication.